# Quantum-Inspired Orthonormal CNN for Energy-Efficient Medical Image Denoising

**Sayantan Dutta**[1]  ID                                    SAYANTAN.DUTTA1@GEHEALTHCARE.COM

[1] *Science and Technology Organization, GE HealthCare, Bangalore 560066, Karnataka, India.*

**Editors:** Accepted for publication at MIDL 2026

## Abstract

Medical imaging modalities (MRI, CT, PET, US) are often degraded by acquisition noise, which obscures subtle anatomical details and compromises diagnostic reliability. Conventional denoising approaches, including spatial filters and deep learning (DL) models, often struggle to balance noise suppression with preservation of fine structures, and state-of-the-art architectures typically incur high computational and energy costs. This work introduces a novel quantum-inspired convolutional neural network (QICNN) that embeds principles of orthonormal basis representation and unitary channel mixing into a compact UNet-style architecture. By constraining convolutional kernels to orthonormal subspaces and enforcing norm-preserving transformations, QICNN eliminates feature redundancy, stabilizes optimization, and maintains energy consistency across layers. Evaluations on real noisy brain MRI datasets show that QICNN achieves superior texture fidelity and lesion conspicuity compared to standard DL models, as evidenced by improvements in GLCM-based metrics and contrast-to-noise ratio. In addition to quality gains, QICNN reduces parameter count by $\sim$93%, inference latency by $\sim$98%, and energy consumption by $\sim$97% relative to transformer-scale denoisers, significantly lowering computational overhead and carbon footprint. These findings highlight the potential of physics-guided design to deliver interpretable, efficient, and clinically robust solutions for medical image restoration.

**Keywords:** Medical image denoising, Quantum-inspired CNN, Orthonormal feature encoding, Unitary transformations, Energy-efficient architecture.

## 1. Introduction

Modern medical imaging modalities (MRI, CT, PET, US) are essential for diagnostics and treatment planning, yet real-world scans suffer from acquisition noise due to hardware limitations, motion, or low-dose protocols (Chi et al., 2025; Martínez-Jiménez et al., 2025; Dutta et al., 2021a). Noise in MRI and CT blurs tissue boundaries, reduces contrast, and obscures small lesions (Talebi and Milanfar, 2014; Lei et al., 2024; Rao and Reddy, 2022; Fan et al., 2020), potentially causing misinterpretation or repeat scans. Robust denoising that preserves anatomical detail is critical for accurate diagnosis (Tapio et al., 2021; Erickson et al., 2017; Yang et al., 2015). However, denoising remains challenging: excessive smoothing erases fine structures, while insufficient filtering leaves residual noise (Patil and Bhosale, 2022).

Classical approaches operate in spatial or transform domains. Spatial filters (Gaussian, median, bilateral) are computationally cheap but oversmooth high-frequency details (Guo et al., 2022). Transform-domain methods (e.g., wavelets) selectively attenuate noise-laden frequencies while preserving low-frequency anatomy, yet rely on handcrafted priors and

degrade under low-SNR (signal-to-noise-ratio) or correlated noise (Do et al., 2022; Dutta et al., 2021b; Langoju et al., 2024). Deep learning (DL) has advanced restoration by learning data-driven priors. CNNs such as DnCNN demonstrated residual learning and batch normalization outperform classical methods for Gaussian denoising (Zhang et al., 2017). Encoder–decoder and residual architectures have been adapted for low-dose CT and MR denoising, e.g., RED-CNN (Chen et al., 2017) and other variants (Kulathilake et al., 2023) for CT and CNN autoencoders for MR (Rahman et al., 2025). Hybrid frameworks combining residual CNNs with dictionary learning or sparsity priors further reduce signal leakage and mitigate Gibbs artifacts in MRI/CT denoising (Rai et al., 2021). Meanwhile, transformer-based denoisers (SwinIR (Liang et al., 2021), Restormer (Ding et al., 2022)) achieve state-of-the-art performance, but their tens of millions of parameters and high carbon footprint make them impractical for real-time clinical deployment (Darestani et al., 2024; Yang et al., 2024). Self-supervised methods (Noise2Noise, Noise2Void) and blind-spot networks reduce reliance on clean targets (Krull et al., 2019; Dutta et al., 2024c), while WISTA-Net integrates wavelet transforms and attention for MR/CT denoising without ground truth (Huang et al., 2024). These trends mark a shift toward deep, attention-driven architectures of increasing complexity to balance noise suppression with structure preservation. Yet conventional DL models still face critical limitations in clinical settings:

- **Information bottleneck:** Encoder–decoder designs downsample features, discarding high-frequency details; skip connections help but subtle textures and small lesions often remain attenuated, which is critical for medical images requiring fine contrast.

- **Heuristic design:** CNNs and transformers stack convolutions and attention blocks empirically, prioritizing performance over principled design, resulting in black-box models with poor interpretability and no assurance of information preservation.

- **Feature redundancy and gradient instability:** Unconstrained kernels often learn correlated filters, wasting capacity and risking vanishing/exploding gradients, requiring extensive normalization. Residual/dense skip connections help but do not enforce any explicit constraint on preserving feature energy throughout the network depth.

- **High computational and energy cost:** State-of-the-art models demand millions of parameters and large floating-point operations (FLOPs), inflating training time, memory, and carbon footprint. For clinical deployment, this raises operational cost and motivates compact, sustainable architectures.

In parallel, quantum-inspired approaches offer a principled alternative by embedding physical constraints into image restoration. Prior work such as DeQuIP formulates denoising as a quantum many-body problem, using Hamiltonian eigenbasis for adaptive orthonormal representation (Dutta et al., 2022a; Floquet et al., 2024), while DIVA unrolls this algorithm into a compact CNN with interpretable modules (Dutta et al., 2024a), extending to multiple medical imaging modalities (Dutta et al., 2022c,b; Dutta and Mamou, 2024, 2025). Building on these advances, this study investigates whether quantum principles—representation in a complete orthonormal basis and evolution via unitary, norm-preserving transformations—can be embedded directly into a fully learned convolutional architecture for medical

Figure 1: Comparative computational footprint of three denoising methods on $512{\times}512$ input. Horizontal bars indicate parameter count, inference time, energy use, and $CO_2$ emissions. QICNN shows the smallest footprint—1.91M parameters, 1.14 ms inference, 0.63J energy, and 0.11g $CO_2$—outperforming RED-CNN and HAIR in efficiency and sustainability for real-time deployment.

image denoising. The proposed design employs convolutional layers whose filters form orthonormal basis and inter-layer transforms constrained to preserve energy. This ensures decorrelated, non-redundant channels, prevents artificial energy loss or amplification, and replaces ad-hoc heuristics with a principled design. This work introduces a novel *Quantum-Inspired Convolutional Neural Network* (QICNN) that integrates these principles into a lightweight, modality-agnostic architecture applicable to CT, MRI, ultrasound, and PET (with preliminary CT results (Appendix D) supporting cross-modality feasibility). The evaluation focuses on real noisy brain MRI, where noise originates from acquisition rather than synthetic perturbations. The key contributions are summarized as follows:

1. **Quantum-inspired orthonormal convolutional architecture:** Convolutional filters are constrained to form orthonormal basis, enabling energy-stable, information-preserving transforms analogous to unitary operators, addressing bottlenecks and redundancy in conventional CNNs.

2. **Physically motivated, compact design:** Architecture guided by quantum principles improves interpretability, and achieves orders-of-magnitude fewer trainable parameters than standard CNN and transformer-based denoisers while maintaining competitive or superior restoration quality (Fig. 1). By embedding physics principles into DL, QICNN represents a paradigm shift toward sustainable AI in medical imaging.

3. **Application to real noisy MRI:** QICNN is validated on clinically acquired noisy brain MRI data. Quantitative (contrast-to-noise ratio, texture descriptors) and qualitative results show superior noise suppression and structure preservation over standard DL baselines, at substantially lower computational and energy cost.

## 2. Methodology

This section outlines the theoretical foundations of QICNN, drawing parallels between quantum principles and their DL interpretations. Each concept is introduced with its *Dirac notation*, physical intuition, and corresponding neural implementation. QICNN adopts two

modules—*Quantum Gate* and *Orthonormal Basis*—that replace standard convolutional filters in the encoder, inspired by Hilbert-space orthonormality and quantum unitary evolution. These modules enforce energy preservation and feature decorrelation, improving stability and interpretability for medical image denoising.

## 2.1. Quantum Gate: Orthonormal Kernels via Unitary Transformations

**Quantum Principle:** A quantum gate is a unitary operator $U$ that transforms a state $|\psi\rangle$ without changing its norm. Unitarity ($U^\dagger U = I$) ensures total probability (energy) conservation. Indeed, a qubit gate rotates or reflects the state on the Bloch sphere without altering magnitude. Complex gates may entangle qubits, but fundamentally implement norm-preserving linear operations (Dutta et al., 2024b).

Modeling image features as quantum states, each channel vector $|\psi\rangle$ represents amplitudes of a multi-level system. A unitary transformation rotates this state in Hilbert space, redistributing information across channels while preserving energy (without amplification or attenuation)—analogous to phase rotations in wave mechanics.

**Deep Learning Translation:** The Quantum Gate module implements a $1 \times 1$ convolution with weight matrix $W \in \mathbb{R}^{n \times n}$ constrained to be orthonormal ($W^\mathsf{T} W = I_n$). This acts like a quantum gate operating on the "quantum state" represented by the feature vector. For $n$ channels, if $f_{in}(p) \in \mathbb{R}^n$ is the input feature vector at pixel $p$, then output feature vector $f_{out}(p) = W f_{in}(p)$, where $W$ performs a learned orthonormal transformation (unitary rotation) mixing channels without altering energy of the input feature vectors $f_{in}(p)$. During training, orthonormality is enforced via QR decomposition: an unconstrained matrix is factorized as $W = QR$, retaining only the orthonormal factor $Q$ and absorbing $R$ into scaling if needed. Since $W$ is orthonormal, its inverse is $W^\mathsf{T}$, making the operation locally invertible. A ReLU follows each unitary layer to introduce non-linearity; batch normalization is omitted as orthonormality stabilizes activations (keeping variance consistent: $E[f_{out} f_{out}^T] = W E[f_{in} f_{in}^T] W^T$), reducing overhead and improving interpretability.

This is essentially a norm-preserving channel mixing layer. In contrast, standard CNNs use $1 \times 1$ convolutions for arbitrary linear combinations that may amplify or shrink activations (Gavrikov and Keuper, 2023). Each Quantum Gate restricts mixing to rotations—expressive enough to reorient features without rescaling—helping maintain balanced feature distributions. Since $W$ is orthonormal, the operation is *invertible*, akin to a local normalizing flow-like characteristic. Although global invertibility is not enforced due to other nonlinear layers, each Quantum Gate can be reversed by its transpose $W^T = W^{-1}$, ensuring no information is lost when passing through it.

**Implications for Medical Image Denoising:** Unitary channel mixing preserves feature energy and avoids noise amplification, enabling stable convergence and interpretability. In MRI, where coil sensitivities and phase variations can mix signals across channels, a norm-preserving rotation offers robustness to phase noise and coil-dependent artifacts. In CT or X-ray, energy-conserving mixing helps suppress streak artifacts by distributing structured noise across independent directions. Because the weights are constrained, the network is less prone to overfitting and maintains stability when trained on small clinical datasets.

## 2.2. Orthonormal Basis: Feature Map Orthonormalization

**Quantum Principle:** In quantum mechanics, any state $|\psi\rangle$ in Hilbert space can be expanded using a complete set of orthonormal basis vectors $|\phi_i\rangle$ satisfying $\langle\phi_i|\phi_j\rangle = \delta_{ij}$. These eigenfunctions, often solutions to operators like the Schrödinger Hamiltonian (Dutta et al., 2021b), represent mutually exclusive modes such as energy levels or vibrations (Hashemi et al., 2025). Applied to imaging, an adaptive orthonormal basis decorrelates image components, separating signal from noise. Feature maps from a convolutional layer can be viewed as samples of a multivariate function; decomposing them into orthonormal components is analogous to projecting a quantum state onto orthogonal modes. Each mode captures a distinct spatial pattern, and the full set spans the feature space without redundancy.

**Deep Learning Translation:** The Orthonormal Basis Layer explicitly orthonormalizes feature maps from a convolutional layer, thereby projecting the feature representation into an orthogonal basis space. Unlike standard CNNs where maps may be correlated, this module ensures channels remain uncorrelated and energy-balanced throughout the network. Let $X \in \mathbb{R}^{n \times h \times w}$ be the output with $n$ channels (same as input channels due to orthonormal filter bank) and spatial size $h \times w$. Flattening yields $X' \in \mathbb{R}^{n \times (hw)}$. To orthogonalize the channels, singular value decomposition (SVD) is applied: $X' = V\Sigma\bar{V}^T$, where $V \in \mathbb{R}^{n \times n}$ is orthonormal matrix representing basis directions across channels, $\Sigma$ diagonal with singular values, and $\bar{V}^\mathsf{T}$ encodes spatial patterns. Orthonormalized feature maps are reconstructed as $X_{\mathrm{orth}} = V\Sigma$, preserving channel orthogonality and energy while discarding $\bar{V}^\mathsf{T}$ to maintain spatial layout. This can be applied per mini-batch or approximated in deeper layers to reduce computation. Note that orthonormalization affects feature maps only; convolutional kernels remain unconstrained in this block. Furthermore, to keep the decomposition efficient, a channel-only SVD is applied to an $n \times (hw)$ matrix (with $n = 96$, for example). Because the channel dimension is small, the dominant computational term remains lightweight. In practice, profiling on GPU shows that this operation accounts for less than 9% of per-batch training compute time. Furthermore, orthonormalization is performed once per mini-batch, reducing the number of SVD calls proportionally to the batch size. Note that the SVD is not used during inference, so no runtime overhead is incurred.

Enforcing orthonormality across feature maps yields several notable advantages. First, it eliminates redundancy: by ensuring each channel is linearly independent, the network captures distinct signal components rather than overlapping patterns. This allows for more efficient use of model capacity and mitigates feature duplication. Second, it preserves energy balance across channels; because the transformation is norm-preserving, no activation disproportionately dominates or diminishes, helping maintain stable gradient flow. Third, orthonormality acts as an implicit regularizer: it discourages the network from memorizing spurious patterns and encourages decorrelation of activations, which can improve generalization. Finally, using a complete orthonormal basis ensures that the input signal is fully represented—if one basis vector cannot capture a particular pattern, another orthogonal vector will—so no relevant information is discarded. Collectively, these properties promote interpretable feature representations and more robust optimization.

**Implications for Medical Image Denoising:** For medical denoising, these advantages translate into tangible benefits. In MRI, orthonormal maps decorrelate coil sensitivities and suppress system noise, preserving subtle anatomical textures critical for diagnosis.

In CT, they distribute structured artifacts—such as streaks or rings—across independent modes, reducing coherence and aiding removal. Balanced energy prevents oversmoothing of finer but diagnostically relevant details and ensures stability under low-SNR conditions. The complete-basis property avoids bottlenecks in latent representation, mitigating detail loss. Aligning feature space with orthogonal directions improves signal-noise separation, enhancing robustness and consistency across modalities.

### 2.3. Overall Architecture: UNet Implementation

QICNN adopts a UNet-style encoder–decoder without downsampling, maintaining constant feature dimensions. The encoder alternates between Quantum Gate modules (unitary kernels) and Orthonormal Basis layers (feature orthogonalization), replacing conventional convolutional blocks. The decoder employs standard deconvolutions with skip connections, where orthonormal feature maps from the encoder are passed directly to corresponding decoder blocks to retain spatial and structural details (see Fig. 6 in Appendix A).

Beyond preserving information, the built-in orthonormality improves optimization landscape, making gradients more predictable and reducing sensitivity to initialization. Combining unitary kernel weights with orthonormal feature maps creates a synergistic effect: transformations remain energy-preserving while activations become decorrelated and balanced across channels. This dual-level orthonormality promotes smoother training dynamics, better generalization, and interpretable representations, yielding a compact yet powerful denoising network suited for diverse medical imaging modalities.

## 3. Experimental Setup and Results

QICNN performance is evaluated against leading architectures, focusing on clinical MRI denoising quality and efficiency. Assessment prioritizes clinically relevant metrics—texture fidelity and lesion conspicuity—using *gray-level co-occurrence matrix* (GLCM) descriptors and *contrast-to-noise ratio* (CNR), with results linked to QICNN's core design principles—unitary channel mixing and orthonormal feature coding—to highlight their impact on image quality and computational footprint. Details of how inference time, energy usage, $CO_2$ emissions, FLOPs, and memory metrics are computed are provided in Appendix B for clarity and reproducibility.

### 3.1. Training Protocol and Evaluation Datasets

The proposed QICNN was trained on $128 \times 128$ patches extracted from the Flickr dataset, using 2000 samples for training and 500 for validation. Training employed the Adam optimizer with an initial learning rate of $1 \times 10^{-4}$ (decayed by 0.99 per epoch), a batch size of 64, and 500 epochs. All convolutions used $5 \times 5$ kernels with 96 channels. The model was implemented in PyTorch and trained on an NVIDIA RTX 6000 Ada GPU cluster. The orthonormalization step uses channel-only, mini-batch grouped SVD and is applied exclusively during training, and no SVD is executed during inference, ensuring that efficiency metrics remain unaffected. Note that this training choice leverages the architecture's orthonormality and unitary constraints, which enforce data-agnostic feature transformations

and therefore do not rely on modality-specific textures, supporting the suitability of natural-image supervision for this architecture.

For clinical validation, QICNN was tested on brain MR images drawn from three publicly available repositories: the TCGA-GBM (Scarpace et al., 2016) and TCGA-LGG (Pedano et al., 2016) collections hosted by The Cancer Imaging Archive, and the IXI dataset (IXI dataset, 2024) consisting of healthy volunteer scans. These datasets encompass both pathological and normal cases, offering broad anatomical and contrast variability. Aggregated statistics are reported across 20 representative studies, using two predefined regions of interest (ROIs) per case. Baselines include standard UNet (Gurrola-Ramos et al., 2021) (without quantum-inspired components), RED-CNN (Chen et al., 2017), and HAIR (Cao et al., 2024), trained under the same regime with comparable channel widths for fair comparison. Efficiency metrics were measured on 512×512 inputs during single-image inference.

### 3.2. Results and Evaluation

Fig. 2 compares denoised outputs for multiple cases (zoomed ROIs highlighted). As expected, acquired MR images (left column) exhibit strong background noise and speckle that obscure low-contrast tissue boundaries. UNet without quantum-inspired components (column 2) suppresses noise without any artifacts, but introduces noticeable blurring and loss of fine details (yellow arrows). RED-CNN (column 3) removes high-frequency noise more aggressively yet often introduces localized distortions near tissue interfaces. HAIR and QICNN both preserve sharper edges and subtle textures; however, QICNN consistently reveals clearer low-contrast structures (e.g., transitions at gray–white matter interfaces; yellow arrows), indicating effective speckle suppression without erasing diagnostically relevant detail. Additional examples appear in Appendix C.

Texture preservation and lesion conspicuity were quantified using standard GLCM descriptors—*entropy*, *energy*, *homogeneity*, *correlation*, *dissimilarity*, and *angular second moment* (ASM)—and CNR over two ROIs per study. Lower entropy and dissimilarity indicate reduced randomness and fewer abrupt transitions; higher energy, homogeneity, ASM, and correlation reflect more orderly textures and preserved gray-level relationships. CNR complements these descriptors by capturing lesion–background separation. GLCM evaluations used consistent hyperparameters: distance $= 1$, angles $= \{0, \frac{\pi}{4}, \frac{\pi}{2}, \frac{3\pi}{4}\}$, and 256 gray levels.

Fig. 3 reports aggregated means $\pm$ standard deviations across 20 studies. Noisy inputs exhibit high *entropy* and *dissimilarity* alongside low *energy*, *homogeneity*, and CNR, consistent with severe texture disorder. All learning-based methods improve these descriptors. QICNN delivers the most balanced outcome, achieving the *highest energy* (0.152±0.056), *homogeneity* (0.613±0.092), and CNR (6.283±2.019) together with the *lowest dissimilarity* (4.355±0.673). UNet and HAIR follow closely (*homogeneity* ∼0.60–0.61, CNR ∼5.72–5.77), while RED-CNN yields intermediate gains (*energy* 0.125±0.035, CNR 5.342±1.545). *Correlation* values near ∼0.97 for all trained models (vs. 0.803±0.087 in the noisy baseline) confirm structural relationships are preserved after denoising. These trends substantiate clinically meaningful noise suppression *without* erosion of important anatomical cues. To assess the statistical reliability of the observed differences, dependent paired t-tests were conducted across all 20 studies. QICNN achieves statistically significant improvements over the noisy input for all GLCM descriptors and CNR ($p < 0.05$, Fig. 3). In addition, QICNN

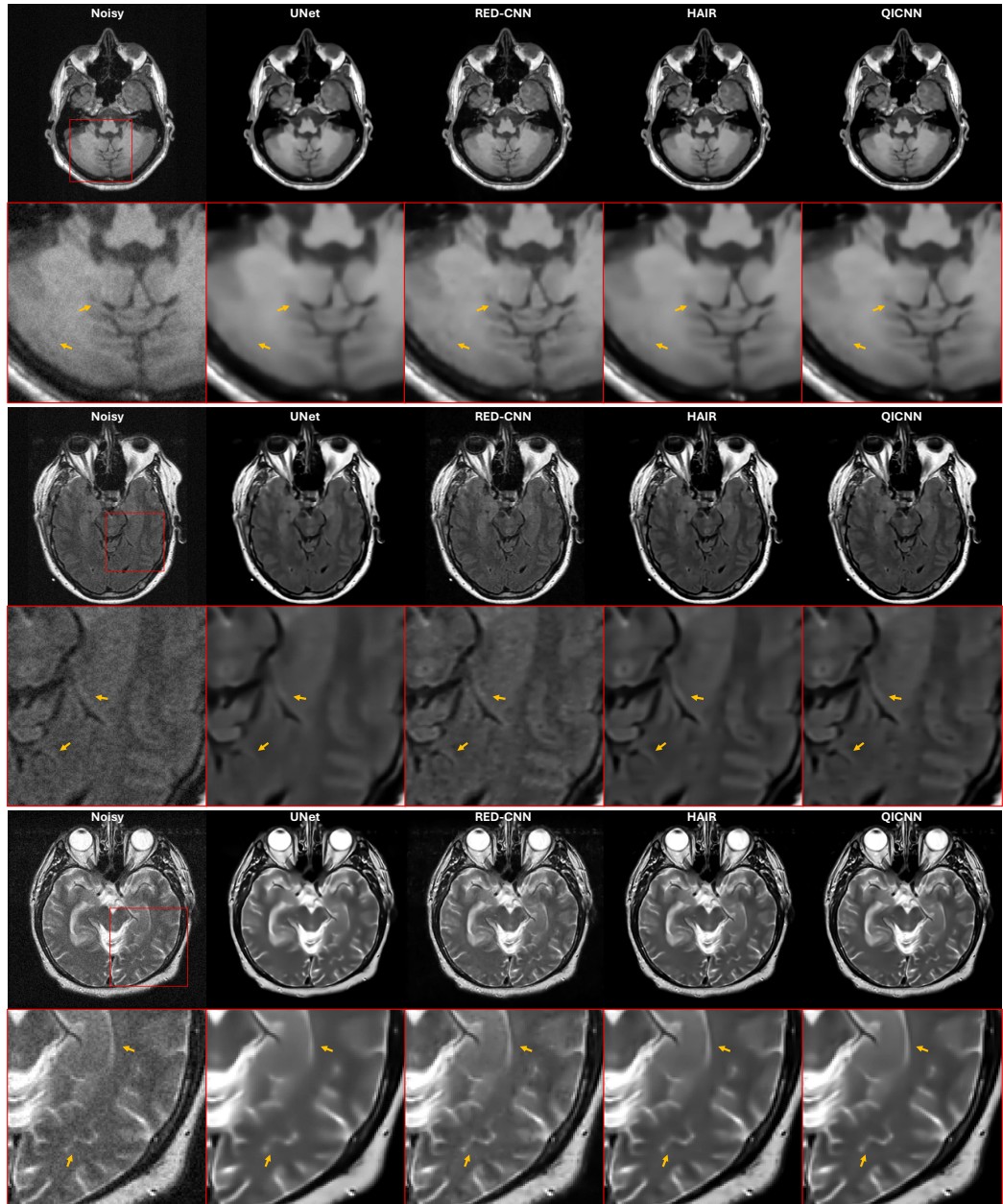

Figure 2: Qualitative comparison across methods (examples with zoomed ROIs). QICNN preserves fine textures and low-contrast boundaries more consistently than UNet (without quantum-inspired components) and RED-CNN, and rivals/surpasses HAIR on several cases. More visual images are available in Fig. 7 in Appendix C.

demonstrates significant gains over RED-CNN on several key descriptors and over UNet on selected metrics. As anticipated from the close quantitative values, differences relative to the recent HAIR model were not statistically significant on this dataset. These findings

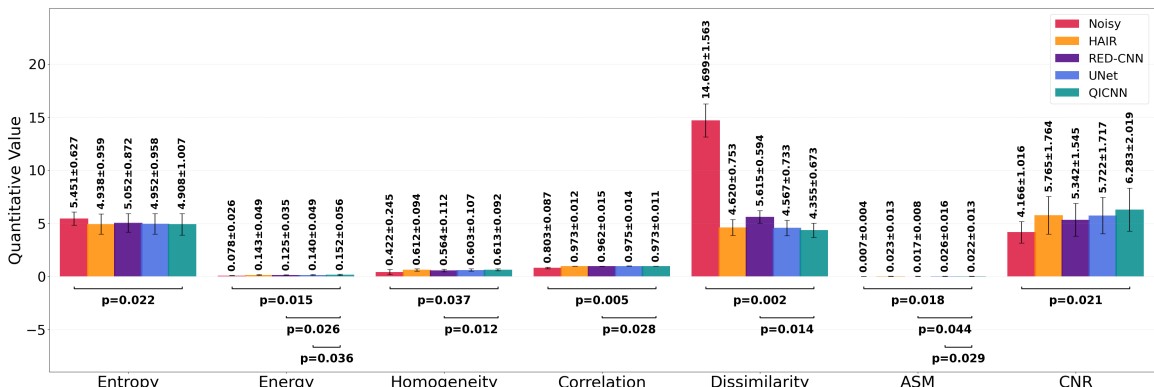

Figure 3: Quantitative texture analysis using GLCM metrics (*entropy, energy, homogeneity, correlation, dissimilarity, ASM*) and CNR across 20 brain MRI cases, comparing noisy input, HAIR, RED-CNN, UNet (without quantum-inspired components), and QICNN; error bars indicate *std* across ROIs. QICNN achieves the best overall balance—higher texture order (*energy, homogeneity, ASM, correlation*), lower disorder (*entropy, dissimilarity*), and superior lesion conspicuity (CNR). Statistically significant differences (paired t-tests, $p < 0.05$) are indicated by horizontal bars with the corresponding p-values.

confirm that the performance gains reported for QICNN are statistically reliable and not attributable to random variation across cases.

As shown in Fig. 1, QICNN achieves superior computational efficiency compared to RED-CNN and HAIR. With only 1.91M parameters, QICNN delivers an inference time of 1.14 ms, consuming 0.63 J of energy and emitting 0.11 g $CO_2$ per inference. In contrast, RED-CNN and HAIR require 18.49M and 28.56M parameters respectively, with significantly higher latency (5.68 ms and 82.15 ms) and energy footprints (2.15 J and 24.23 J). These results underscore QICNN's ability to combine rapid inference with sustainability, making it highly suitable for real-time and resource-constrained clinical environments.

To further substantiate the quantitative advantages of the proposed architecture, a controlled phantom study with available clean ground-truth image was conducted, enabling the use of full-reference fidelity metrics. Under identical noise conditions, peak signal-to-noise ratio (PSNR) and structural similarity index (SSIM) were computed for QICNN, HAIR, UNet, and RED-CNN. Across all phantom cases, QICNN achieves consistently higher PSNR and SSIM values, reflecting superior preservation of fine underlying structures, improved edge integrity, and more faithful global contrast reconstruction (see Fig. 4). These full-reference findings align closely with the clinical MRI results, further reinforcing the efficiency–accuracy advantage conferred by the quantum-inspired design. Notably, QICNN demonstrates PSNR gains of approximately 1.38–2.11 dB and SSIM improvements of 0.14–0.22% relative to the recent HAIR benchmark, highlighting the model's robustness across both real and synthetic evaluation settings.

Finally, ablation studies (Fig. 5) validate QICNN's core constraints. Singular values from Quantum Gate kernels remain consistently at 1 (Fig. 5(a)), confirming norm preser-

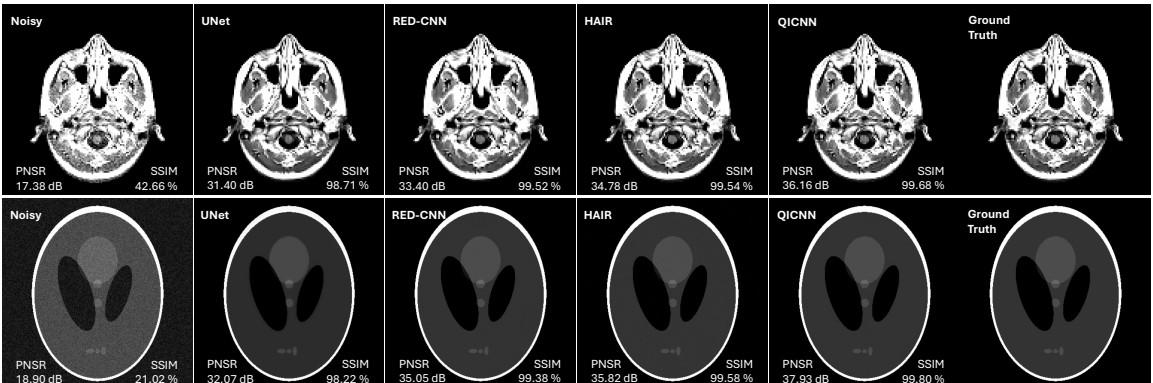

Figure 4: Phantom-based denoising comparison between baseline UNet, RED-CNN, HAIR and the proposed QICNN. QICNN achieves higher PSNR/SSIM and preserves fine structural details and contrast more faithfully.

vation through unitary transformations, ensuring no artificial amplification or attenuation of feature energy. Likewise, Fig. 5(b) shows strictly diagonal inner-product matrices for feature maps, verifying orthonormality. Together, these findings demonstrate that QICNN maintains energy stability and decorrelated representations throughout the network.

## 4. Discussion

The qualitative (Fig. 2) and quantitative (Fig 3) comparisons collectively demonstrate the superiority of the proposed QICNN over conventional DL denoisers. Visual inspection reveals that QICNN consistently suppresses noise while preserving fine anatomical details and low-contrast boundaries—features often blurred by UNet and distorted by RED-CNN. HAIR performs competitively, yet QICNN achieves clearer delineation of subtle tissue interfaces, validating its ability to balance aggressive noise removal with structural fidelity.

Beyond visual performance, QICNN delivers substantial efficiency gains that amplify its practical significance (Fig. 1). Compared to HAIR, it reduces parameter count by 93.3% (1.91M vs. 28.56M), latency by 98.6% (1.14 ms vs. 82.15 ms), energy consumption by 97.4% (0.63 J vs. 24.23 J), and $CO_2$ emissions by 97.5%. Against RED-CNN, reductions are 89.7% in parameters, 79.9% in latency, and 70.7% in energy footprint. These improvements underscore QICNN's ability to combine high-quality restoration with sustainability—increasingly critical requirement for large-scale clinical deployment and edge-based imaging systems.

Quantitative analysis corroborates these observations (Fig 3). Across 20 representative brain MRI cases, QICNN attains the highest *energy* (0.152±0.056) and *homogeneity* (0.613±0.092), alongside the lowest *dissimilarity* (4.355±0.673), indicating that its outputs exhibit the most orderly and uniform texture patterns. The reduction in *entropy* relative to the noisy baseline confirms effective suppression of stochastic intensity fluctuations, while the elevated *correlation* ($\approx 0.973$) ensures that spatial relationships between neighboring pixels remain intact. Importantly, QICNN delivers the highest mean CNR (6.28±2.02), surpassing HAIR (5.77±1.76), RED-CNN (5.34±1.55), and UNet (5.72±1.72). These gains

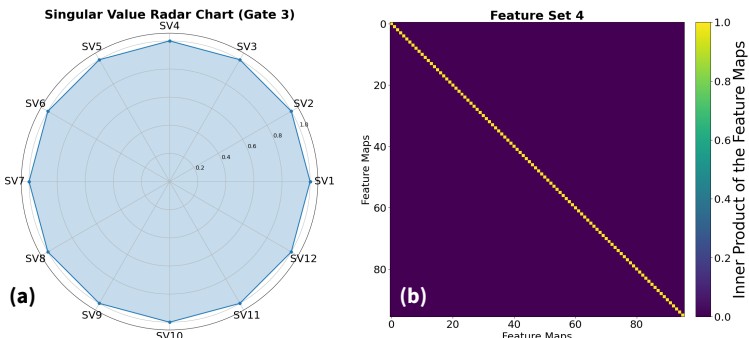

Figure 5: Ablation studies validating QICNN's quantum-inspired constraints. (a) Energy preservation in Quantum Gate 3: singular values of kernel weights remain at 1 across all tested configurations (twelve shown), confirming unitary transformations and norm conservation. (b) Orthonormality in Feature Set 4: the inner-product matrix of feature maps is strictly diagonal, indicating complete orthogonality and decorrelation among channels.

represent a full two-point increase over the noisy input, translating into improved lesion conspicuity and diagnostic visibility without sacrificing texture fidelity.

The observed improvements directly attributed to QICNN's quantum-inspired architectural principles. By enforcing *unitary channel mixing* in the Quantum Gate modules, the network preserves feature energy across layers, preventing noise amplification and stabilizing optimization under low-SNR conditions. This norm-preserving property ensures that denoising does not come at the cost of attenuating subtle anatomical signals. Simultaneously, the Orthonormal Basis layers decorrelate feature maps, eliminating redundancy and promoting balanced activation energy. This orthogonalization acts as an implicit regularizer, enabling QICNN to capture diverse structural patterns without oversmoothing or overfitting. Without these quantum-inspired components—as in the baseline UNet—the model struggles with information bottlenecks, feature redundancy, and gradient instability, leading to noticeable performance drops in both texture fidelity and lesion contrast. Integrating quantum principles resolves these limitations, creating a feature space that is both information-preserving and energy-stable—drawing inspiration from mathematical analogues of unitary and orthonormal transformations—and explaining why QICNN achieves superior texture coherence and contrast enhancement compared to heuristic CNN or transformer designs.

In summary, QICNN's synergy of visual clarity, metric-driven evidence, and computational sustainability demonstrates that quantum-inspired design is not merely theoretical; it translates into tangible improvements in image quality, diagnostic utility, and deployment feasibility. These findings validate the hypothesis that embedding physics-based constraints into deep architectures can yield interpretable, efficient, and clinically robust solutions for medical image restoration. Ablation studies (Fig. 5) reinforce this: unitary channel mixing ensures norm preservation, while orthonormal feature maps eliminate redundancy, enhancing interpretability and compactness. This integration of theoretical rigor with architectural

design underscores the value of quantum-inspired constraints in advancing robust, efficient medical image restoration systems.

Despite its strong performance, the current study has limitations. First, experiments focus on brain MRI; structured and correlated noise patterns common in CT (e.g., streaks, rings) are not explicitly modeled. Extending QICNN to handle such artifacts will require integrating dual-domain strategies that enforce consistency in both sinogram and image spaces. Second, while orthonormal and unitary constraints improve efficiency and interpretability, they have not yet been explored for more complex restoration tasks such as deblurring, super-resolution, inpainting, or compressed sensing. Future work will investigate embedding these principles into architectures for such tasks. Beyond architectural extensions, future designs may incorporate advanced quantum-inspired properties—such as entanglement, many-body interactions, and localization phenomena—to further enhance robustness and efficiency. These additions could further enhance robustness, improve energy efficiency, and enable principled control over information flow. Finally, scaling evaluations to larger cohorts and diverse modalities will be essential to validate clinical applicability and ensure generalization across imaging conditions. A preliminary qualitative evaluation on CatSim-simulated CT images has been included in Appendix D, providing initial evidence that QICNN can generalize beyond MRI despite being trained on natural-image data. Furthremore, while the present study trains QICNN on structurally diverse natural images (Flickr dataset) to emphasize the modality-agnostic nature of the orthonormality constraints, future work will incorporate modality-specific medical datasets (e.g., CT with streak artifacts, ultrasound with speckle) to further characterize domain adaptation and to conduct broader ablation studies on combined natural-medical training regimes.

## 5. Conclusion

In conclusion, proposed QICNN—a novel quantum-inspired CNN enforcing orthonormality and unitary transformations—achieves superior texture fidelity and lesion conspicuity while reducing parameters by 93% and energy consumption by 97%, validating the power of physics-guided design for medical image restoration. These gains translate into faster, greener, and more reliable imaging workflows—critical for scaling AI in resource-constrained healthcare environments. Looking forward, the proposed framework offers a promising foundation for broader restoration tasks and multi-modality deployment, paving the way for interpretable AI systems grounded in physical laws. This study demonstrates that physics-inspired constraints are not just theoretical—they form the foundation for next-generation medical imaging AI.

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

## Appendix A. Proposed QICNN Architecture

A schematic diagram of the proposed *Quantum-Inspired Convolutional Neural Network* (QICNN) architecture is shown in Fig. 6.

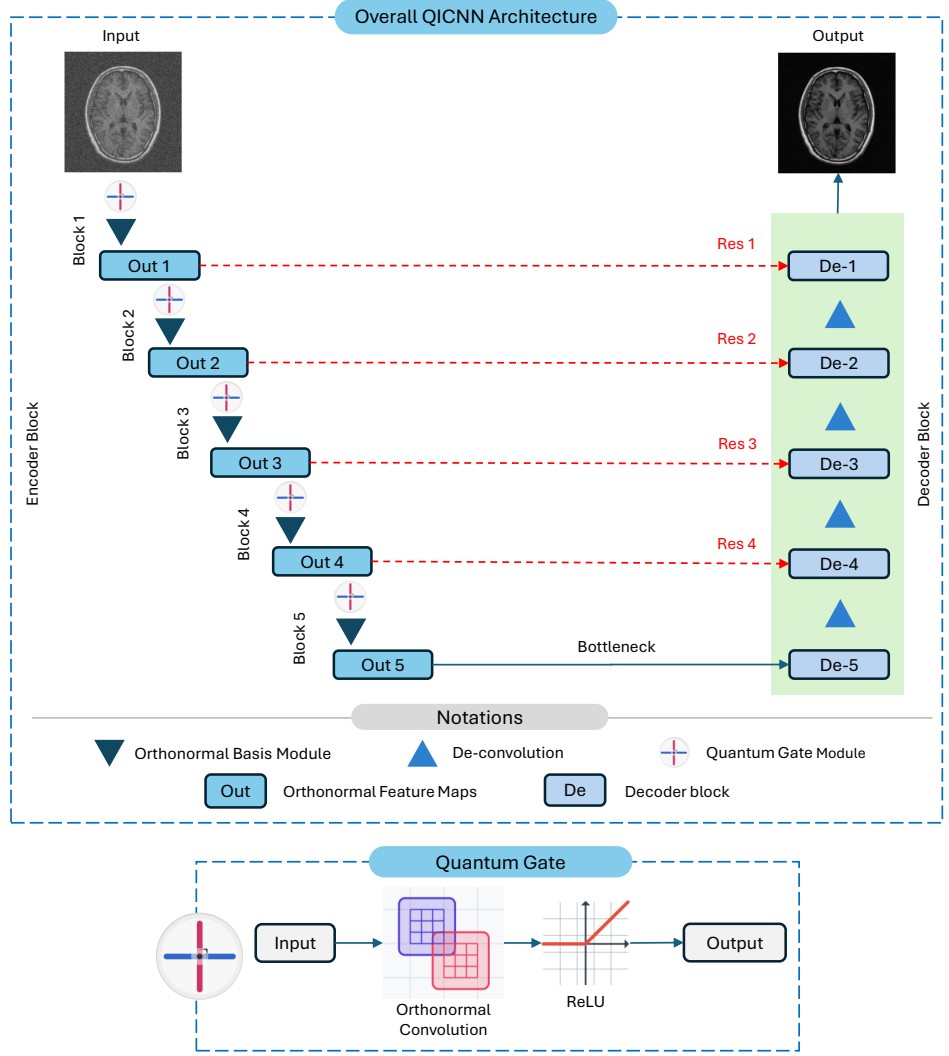

Figure 6: Schematic of the proposed *Quantum-Inspired Convolutional Neural Network* (QICNN). The architecture adopts a UNet-style encoder–decoder design. The encoder alternates between *Quantum Gate* modules (unitary channel mixing with orthonormal kernels) and *Orthonormal Basis* layers (feature map orthogonalization) to enforce energy preservation and feature decorrelation. This module internally uses a lightweight channel-only SVD executed once per mini-batch, ensuring minimal overhead and no impact on inference-time efficiency. Skip connections transfer orthonormal feature maps to corresponding decoder blocks for spatial and structural fidelity, while the decoder uses standard deconvolutions for reconstruction. This design ensures compactness, interpretability, and robust denoising under low-SNR conditions.

## Appendix B. Metric Acquisition Protocol

Inference time is measured as the wall-clock latency of a single forward pass in evaluation mode with gradients disabled. Parameter count and FLOPs are computed using a standardized profiling utility. Energy consumption is obtained by recording instantaneous GPU power draw during the forward pass and integrating it over time to estimate total energy in Joules; $CO_2$ emissions are then derived from this energy using the appropriate regional carbon-intensity factor. Memory usage corresponds to the resident process memory footprint during inference. All models are evaluated under identical hardware and software conditions to ensure fairness.

## Appendix C. More Denoised Images for Qualitative Comparison

Fig. 7 presents additional examples illustrating the qualitative performance of the evaluated denoisers: UNet, RED-CNN, HAIR, and the proposed QICNN. For each case, zoomed-in regions of interest (ROIs) are highlighted with red boxes to facilitate detailed visual assessment.

As expected, the acquired MR images (left column) exhibit severe noise contamination, obscuring fine anatomical structures. The UNet baseline, without quantum-inspired components, (column 2) reduces noise without introducing overt artifacts; however, it produces noticeable blurring and loss of high-frequency details (yellow arrows), which compromises

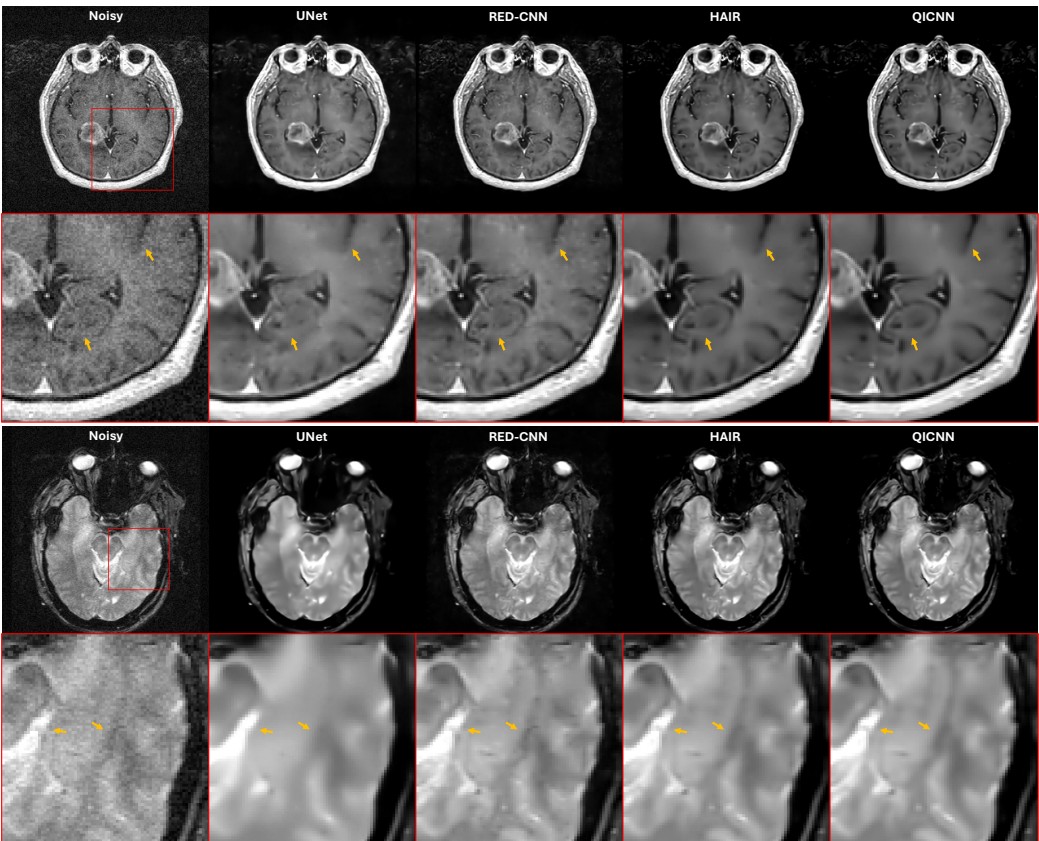

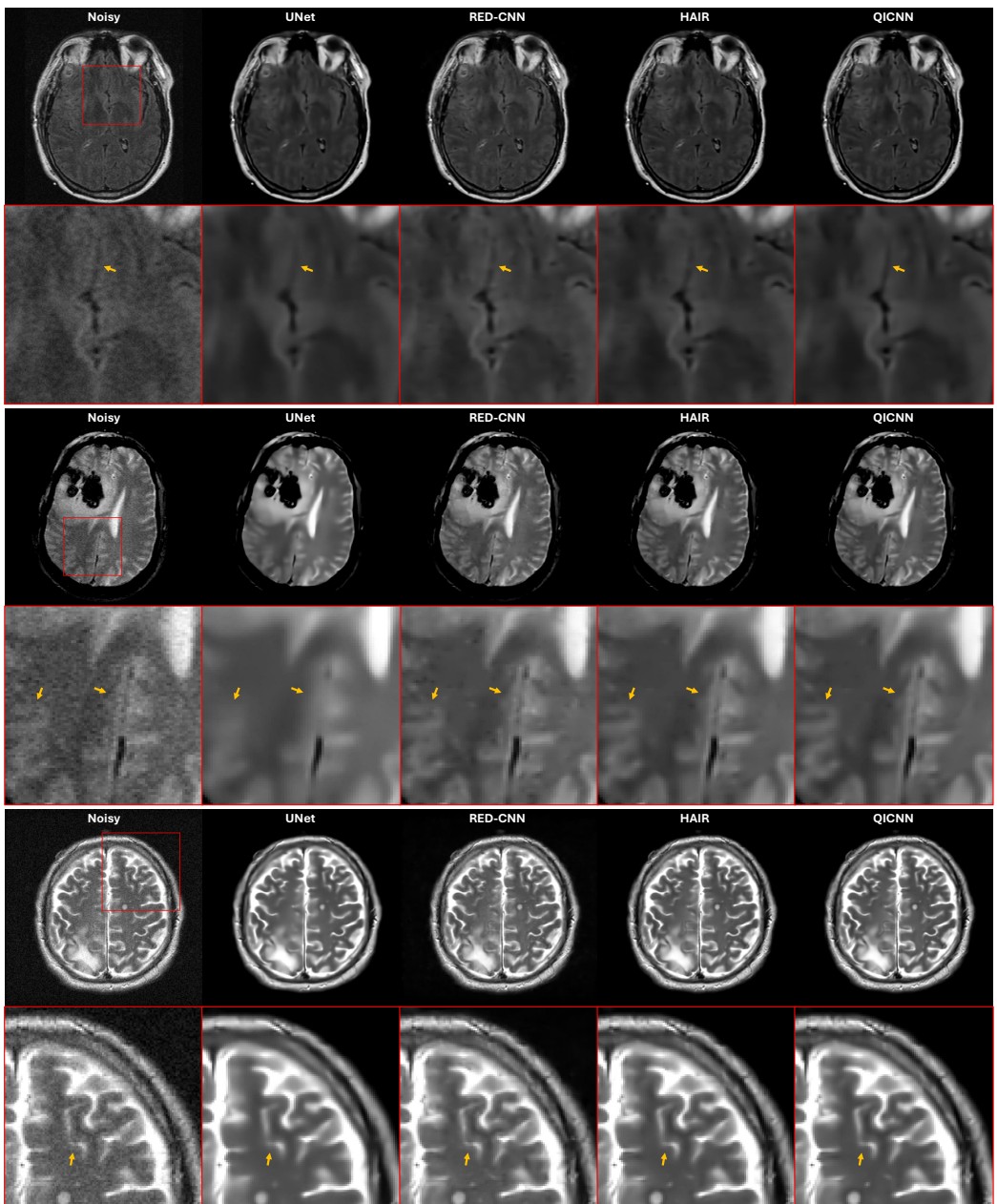

Figure 7: Qualitative comparison across methods (examples with zoomed ROIs). QICNN preserves fine textures and low-contrast boundaries more consistently than UNet (without quantum-inspired components) and RED-CNN, and rivals/surpasses HAIR on several cases.

the visibility of subtle tissue boundaries. RED-CNN (column 3) achieves stronger noise suppression but introduces structural distortions and localized anomalies in complex regions

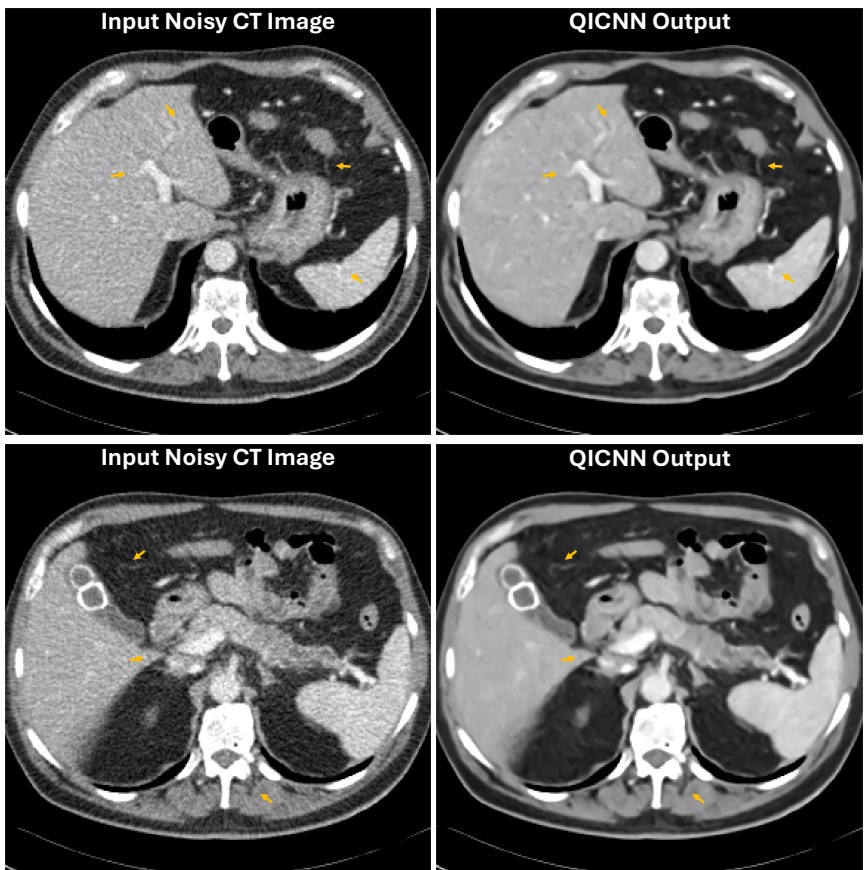

Figure 8: Preliminary CT denoising results using the proposed QICNN on CatSim-simulated low-dose CT images (120 kVp, 200 mA, 1000 detectors, 984 views). QICNN suppresses noise and streak-like artifacts while preserving structural boundaries, despite being trained exclusively on natural images. All images are displayed with [WL 50, WW 300].

(e.g., inter-tissue interfaces), as indicated by yellow arrows. These artifacts can misrepresent anatomical continuity and degrade diagnostic reliability.

In contrast, HAIR (column 4) and QICNN (column 5) deliver the most visually consistent outputs. Both methods substantially attenuate noise while preserving sharp anatomical edges. Notably, QICNN demonstrates superior retention of low-contrast features compared to HAIR, ensuring clearer delineation of soft-tissue transitions and improved visualization of intra-tissue structures. These observations reinforce QICNN's ability to balance aggressive noise removal with structural fidelity, validating its effectiveness for clinical MRI denoising.

## Appendix D. Preliminary CT-Based Evaluation

To provide preliminary evidence supporting the modality-agnostic capabilities of the proposed QICNN architecture, an additional experiment was conducted on computed tomog-

raphy (CT) image denoising. Because realistic CT noise patterns arise from the X-ray photon interaction model and detector geometry, a physics-accurate simulation framework was used to generate controlled low-dose CT measurements. Specifically, clean clinical CT slices were forward-projected using the CatSim simulator under a 120 kVp X-ray source and a tube current of 200 mA, with a fan-beam geometry consisting of 1000 detectors and 984 projection views. The resulting sinograms were then reconstructed to obtain noisy CT images that emulate realistic acquisition conditions.

Although QICNN was trained exclusively on natural-image patches, its orthonormal kernel constraints and unitary channel mixing are designed to learn modality-agnostic transformations that generalize beyond a single imaging domain. To assess this qualitatively, QICNN was applied to the simulated low-dose CT slices without retraining or fine-tuning. Only QICNN outputs are shown here, since directly comparing with other denoising architectures would be confounded by differing training distributions and modality-specific assumptions.

As illustrated in Fig. 8, QICNN effectively suppresses quantum noise and streak-like fluctuations while preserving anatomical boundaries and subtle structural transitions. The denoised outputs appear smooth yet structurally coherent, demonstrating that the orthonormality-driven design confers a degree of cross-modality robustness even in this preliminary setting. These qualitative results serve as an initial indication of QICNN's modality-agnostic behavior, while more comprehensive multi-modality evaluations using real CT datasets will be explored in future work.

