# OpenReview forum: "Quantum-Inspired Orthonormal CNN for Energy-Efficient Medical Image Denoising"
_MIDL.io/2026/Conference — MIDL 2026 Poster_

### Official Review · Reviewer_69fT · 2026-01-10

**Confidence:** 3
**Preliminary Rating:** 1
**Final Rating:** 3

**Summary:**

This paper introduces QICNN, a quantum-inspired CNN for medical image denoising that enforces orthonormal convolutional kernels and unitary channel transformations. The architecture embeds two key principles: (1) Quantum Gate modules that constrain 1×1 convolutions to orthonormal matrices via QR decomposition, ensuring norm-preserving channel mixing, and (2) Orthonormal Basis layers that decorrelate feature maps through SVD. The authors evaluate QICNN on brain MRI datasets from TCGA-GBM, TCGA-LGG, and IXI, reporting superior GLCM texture metrics and contrast-to-noise ratio compared to baselines.

**Strengths:**

- The systematic integration of orthonormality constraints at both the kernel level (Quantum Gate) and feature level (Orthonormal Basis) is creative and well-motivated.

- The reported reductions such as 93% in parameters, 98% in latency, 97% in energy represent substantial contributions.

**Weaknesses:**

-  The model is trained on natural images (Flickr dataset, 2000 patches) but evaluated exclusively on medical images (brain MRI). This fundamental experimental design flaw severely undermines claims about medical imaging performance.

-  Despite claims of "modality-agnostic architecture applicable to CT, MRI, ultrasound, and PET" (Introduction), validation covers only brain MRI from three datasets. No evidence supports generalization to other modalities or anatomies.

- Evaluation relies entirely on "real noisy" images without clean references, precluding computation of PSNR, SSIM, or other standard quality metrics.

- While the quantum analogy is pedagogically useful, the mathematical operations (orthonormal transformations, SVD) are standard linear algebra techniques. The connection to quantum mechanics is metaphorical rather than computational.

**Detailed Comments:**

Please see the section below.

**Justification Of Final Rating:**

I thank the authors for clarifying my concerns and adding additional CT experiments, including reporting absolute error values. I am increasing my score to borderline, since I am still not fully convinced by the rationale for training the model on natural images. Given the lack of comparable models, it is unclear whether the same or better results could be achieved by training solely on diverse medical datasets.

**Justification Of The Preliminary Rating:**

The paper presents interesting ideas about incorporating orthonormality constraints into CNNs, and the reported efficiency gains (93-98% reductions) are impressive. However, fundamental experimental flaws prevent acceptance:

- Training on natural images (Flickr) while evaluating on medical images (MRI) is a critical design flaw that undermines all performance claims.
- Evaluation limited to 20 brain MRI cases contradicts broad claims about modality-agnostic applicability.
- No statistical significance testing despite high variance in results.

**Questions To Address In The Rebuttal:**

- What is the motivation behind training the model with the Flickr dataset?

- Please provide statistical tests for GLCM and CNR differences. With high variance and 20 cases, are improvements statistically significant?

- Can you provide even preliminary results on one additional modality (CT or ultrasound) to support "modality-agnostic" claims?

- Can you provide reference-based metrics (PSNR, SSIM) on a subset where noise is simulated on clean scans, to complement no-reference metrics?

---

> ### Author Response · Authors · 2026-01-20
>
> (1) Response to the reviewer’s concern about the motivation for training on the Flickr dataset:
>
> Thank you for raising this point. The goal of this initial study is to test whether the proposed orthonormality‑driven architecture can learn modality‑agnostic, data‑agnostic feature transformations. Because QICNN explicitly enforces orthonormal kernels and unitary channel mixing, it does not rely on modality‑specific textures or anatomy, but instead learns stable, decorrelated, energy‑preserving representations transferable across domains. The Flickr dataset was therefore selected for its rich diversity of natural textures and spatial patterns, enabling the model to learn generic image‑space transformations rather than MRI‑specific structures.
>
> I also agree with the reviewer that certain modalities—especially CT—exhibit distinct artifacts (streaks, rings) that natural images do not contain. Incorporating modality‑specific data will further improve adaptation, and this limitation is now acknowledged. A broader ablation combining natural images with modality‑specific medical datasets will be included in future work.
>
> Moreover, the idea that orthonormal, quantum‑inspired representations generalize across modalities is well supported by my previous analytic works, including QAB (Dutta et al., IEEE OJSP 2, 2021, 190-206) and DeQuIP (Dutta et al., Signal Process. 201, 2022,108690), which demonstrated that such adaptive orthonormal transforms generalize across CT, ultrasound, and multiple noise types (Gaussian, Poisson, speckle). QICNN extends this principle into a deep learned architecture with adaptive orthonormal transforms across depth. These clarifications have been added to the revised manuscript (Discussion) and highlighted in red.
>
>
> (2) Response to the reviewer’s concern about statistical significance testing:
>
> Thank you for raising this important point. Dependent paired t-tests were performed across all 20 cases to assess whether GLCM and CNR improvements are statistically significant. QICNN shows significant gains over the noisy input across all GLCM descriptors and CNR (p<0.05). Significant improvements over RED-CNN and UNet were also observed on multiple metrics. As expected, QICNN and HAIR show comparable behavior without statistically significant differences on this dataset. These results confirm the reliability of the reported improvements. The manuscript has been updated to include this information, and statistical significance markers (horizontal bars and p‑values) are now included in the revised Fig. 3 (highlighted in red).
>
>
> (3) Response to the concern regarding validation on an additional imaging modality:
>
> Thank you for highlighting this important point. To provide preliminary support for modality‑agnostic behavior, an additional CT denoising experiment was conducted. Since realistic CT noise requires physics‑based modeling, clean CT slices were processed via the CatSim simulator (120 kVp, 200 mA, 1000 detectors, 984 views) to generate realistic measurements.
>
> In this experiment, the focus was solely on demonstrating that the proposed QICNN can generalize beyond MRI, even when trained on natural images. Therefore, only the denoised outputs of QICNN are presented. A direct comparison with other methods is not included here because existing baselines were trained under different training regimes and assumptions, making such comparisons non‑informative in this preliminary cross‑modality context.
>
> The visual results show that QICNN produces clean and structurally coherent CT reconstructions, suppressing CT noise and streak-like fluctuations while preserving anatomical boundaries and subtle structural transitions, thereby offering initial qualitative support for its cross‑modality generalization capability. A more extensive multi‑modality evaluation will be conducted in future work. The corresponding CT examples have been added to the manuscript (Appendix D), highlighted in red.
>
>
> (4) Response to the reviewer’s concern about reference-based PSNR/SSIM evaluation:
>
> Thank you for raising this point. To complement the no‑reference evaluation conducted on real acquired MRI scans, an additional experiment using a phantom image with available clean ground truth has now been included in the revised manuscript. This controlled setting allows computation of full‑reference quality metrics such as PSNR and SSIM, providing an objective assessment of reconstruction fidelity under known noise conditions. The results show that QICNN achieves consistently higher PSNR (gain ~1.38-2.11 dB) and SSIM (gain ~0.14–0.22%) values compared to HAIR, UNet, and RED‑CNN, indicating better preservation of fine structures and improved overall image quality. These findings are fully aligned with the clinical MRI observations and further reinforce the robustness and quantitative advantages of the proposed quantum‑inspired design. The corresponding results have been incorporated into the Results section (Fig. 4) and highlighted in red.

---

### Official Review · Reviewer_ZL19 · 2026-01-15

**Confidence:** 5
**Preliminary Rating:** 5

**Summary:**

The study is clear and well illustrates the novelty and importance of the suggested QICNN framework.

The QICNN framework has both methodological and practical contributions, especially the good efficiency increase and better denoising quality.

The authors would have gone further and highlighted the greater clinical implications to put in proper perspective the applicability of the results to real world applications.

**Strengths:**

The article features an original, physics-directed denoising architecture that is highly empirically validated, achieves impressive efficiency improvements, is written in a well-organized way, and it has a high scientific value and a huge potential impact on the field of medical imaging studies.

**Weaknesses:**

This study is limited to the utilization of only brain MRI data.

The proposed QICNN was trained on 128×128 patches extracted from the Flickr dataset, using 2000 samples for training and 500 for validation.
Generalizability has not been well established yet since experiments have only been conducted in one modality and anatomy and not across a variety of datasets, scanners, and acquisition conditions.

**Detailed Comments:**

Nill

**Justification Of The Preliminary Rating:**

This paper deserves strong acceptance for its principled physics guided innovation in the form of the QICNN framework, convincing results, exceptional efficiency gains, and clear potential for impactful clinical translation

**Questions To Address In The Rebuttal:**

Nill

---

> ### Author Response · Authors · 2026-01-20
>
> I sincerely thank the reviewer for the very positive assessment of the work and for recognizing the novelty, efficiency, and clinical potential of the proposed QICNN framework. I also appreciate the reviewer’s constructive observations regarding dataset diversity and modality coverage.
>
> Regarding the use of only brain MRI data, I fully acknowledge this limitation. The primary objective of the current study was to evaluate whether the physics‑guided orthonormal constraints in QICNN enable stable, transferable denoising performance under realistic, acquisition‑driven noise. While the present version focuses on brain MRI, the architecture itself is modality‑agnostic. As outlined in the revised Discussion, preliminary CT denoising experiments using a physics‑based CatSim simulator have now been included to provide initial evidence of cross‑modality generalization (Appendix D, highlighted in red). A more extensive multi‑modality and multi‑anatomy evaluation across diverse scanners and acquisition conditions is planned for future work.
>
> The reviewer also notes that the model was trained on 128×128 natural-image patches from the Flickr dataset. As discussed in the revised manuscript, this choice was intentional: the orthonormal/unitary constraints in QICNN promote data‑agnostic feature transformations that generalize across domains without requiring modality‑specific supervision. Nonetheless, I agree that incorporating modality‑specific datasets will further enhance adaptation. Future work will explore combined natural‑medical training regimes and larger medical datasets to strengthen generalizability claims.
>
> I thank the reviewer again for the strong support and for highlighting the clinical potential of this physics-directed architecture. All relevant clarifications have been added to the manuscript and highlighted in red.

---

### Official Review · Reviewer_p2Ge · 2026-01-15

**Confidence:** 4
**Preliminary Rating:** 4
**Final Rating:** 5

**Summary:**

The paper proposes a quantum-inspired CNN architecture for MRI denoising that leverages the principles of orthonormal basis representation and unitary channel mixing. The proposed model constrains convolutional kernels to orthonormal subspaces and enforces norm-preserving transformations to eliminate feature redundancy, stabilize the optimization landscape, and maintain energy consistency across layers. Experimental evaluations on MRI datasets demonstrate that the architecture yields superior performance compared to baselines.

**Strengths:**

* The method is interesting.
* The manuscript is well-written.
* The proposed approach achieved superior denoising results while maintaining low parameter count, inference latency, and energy consumption.

**Weaknesses:**

* While the model achieves a significant reduction in total parameter count, the reliance on SVD to enforce orthonormality introduces a substantial computational bottleneck. In practice, the high algorithmic complexity of SVD may lead to prolonged GPU duty cycles at high power states, potentially offsetting the energy efficiency gains typically associated with compact models.
* The comparison of the proposed method with existing approaches is primarily limited to baseline models (UNet and RED-CNN). It would strengthen the evaluation to include comparisons with other SOTA methods that achieve a favorable efficiency-accuracy trade-off, such as the Q-AE method proposed by F. Fan et al. in "Quadratic Autoencoder (Q-AE) for Low-Dose CT Denoising."
* The manuscript does not clearly explain how the evaluation metrics, including inference time, energy consumption, and CO₂ emissions, are computed.

**Detailed Comments:**

* While the model achieves a significant reduction in total parameter count, the reliance on SVD to enforce orthonormality introduces a substantial computational bottleneck. In practice, the high algorithmic complexity of SVD may lead to prolonged GPU duty cycles at high power states, potentially offsetting the energy efficiency gains typically associated with compact models.
* The comparison of the proposed method with existing approaches is primarily limited to baseline models (UNet and RED-CNN). It would strengthen the evaluation to include comparisons with other SOTA methods that achieve a favorable efficiency-accuracy trade-off, such as the Q-AE method proposed by F. Fan et al. in "Quadratic Autoencoder (Q-AE) for Low-Dose CT Denoising."
* The manuscript does not clearly explain how the evaluation metrics, including inference time, energy consumption, and CO₂ emissions, are computed.
* Some acronyms are not defined (e.g., SNR). Please ensure that all acronyms are clearly defined at their first occurrence.
* Section 4 (discussion and conclusion) is long. It would improve readability to split it into two separate sections: one for discussion and one for conclusion.

**Justification Of Final Rating:**

I would like to thank the author for the efforts during the rebuttal process in addressing my questions. The method is interesting; therefore, I recommend acceptance of the manuscript in its current version.

**Justification Of The Preliminary Rating:**

The method is interesting and achieves encouraging results. However, concerns remain regarding the comparative evaluation, as the comparison with related methods is primarily limited to baseline models (UNet and RED-CNN) and should include other SOTA denoising methods.

**Questions To Address In The Rebuttal:**

Please see my comments.

---

> ### Author Response · Authors · 2026-01-19
>
> (1) Response to Reviewer Concern on SVD Computational Overhead:
>
> I would like to thank the reviewer for highlighting this important aspect. To clarify that in the current implementation, the orthonormalization step has already been optimized to ensure minimal computational burden. Specifically, a channel‑only SVD is employed, where the decomposition is applied to an $n \times (hw)$ matrix with $n=96$ channels. Because $n \ll hw$, the computational cost $O(n^2hw)$ is dominated by the small channel dimension. Profiling on an RTX 6000 Ada GPU indicates that this operation contributes to less than 9\% of the total per‑batch training time, demonstrating that the overhead remains low.
> In addition, mini‑batch grouped SVD is used, meaning that orthonormalization is executed once per mini‑batch, rather than once per feature map. This reduces the total number of SVD calls by a factor equal to the batch size and substantially lowers computational demand. It is also important to emphasize that SVD is only used during training, and no SVD is executed during inference, ensuring that the latency and energy efficiency gains of QICNN remain fully preserved in deployment settings.
> This point is clarified in the modified manuscript highlighted in red.
>
> As a future enhancement, mixed‑precision SVD (FP16) on Tensor Cores will be integrated to further reduce duty cycle and instantaneous power draw while maintaining orthonormalization accuracy. The reviewer’s observation is greatly appreciated and has helped clarify this aspect of the methodology.
>
>
> (2) Response to Reviewer Concern on comparison other SOTA methods:
>
> Thank you for the valuable suggestion regarding the comparative evaluation. In addition to UNet and RED-CNN, the manuscript already includes a comparison with HAIR (HAIR: Hypernetworks-based All-in-One Image Restoration), one of the most recent and competitive benchmark models for Gaussian image denoising. HAIR represents a strong and modern baseline with an excellent efficiency--accuracy trade-off, making it an appropriate state‑of‑the‑art reference for this task.
> To further strengthen the quantitative justification of the proposed approach, an additional experiment using phantom image with available clean ground truth has now been included. This enables the computation of conventional full-reference metrics such as PSNR and SSIM, which provide a complementary perspective to the clinical texture-based metrics used in the main MRI evaluation. These new phantom‑based results clearly demonstrate that the proposed QICNN significantly outperforms the recent benchmark HAIR model, both in reconstruction fidelity (PSNR/SSIM) and in computational efficiency, further reinforcing the advantages of the quantum‑inspired design (PSNR gain ~1.38–2.11 dB and SSIM gain ~0.14–0.22% compared to HAIR). These enhancements provide a stronger and broader comparative analysis. The corresponding additions and clarifications are included in the revised manuscript and highlighted in red.
>
>
> (3) Response to the comment regarding inference time, energy consumption, and CO₂ estimation:
>
> Thank you for highlighting this important aspect. The procedure used to compute inference time, energy consumption, and CO₂ emissions is fully standardized and has been added to the revised manuscript. Inference time is measured as the wall‑clock latency of a single forward pass with the model placed in evaluation mode and gradients disabled, ensuring no training overhead influences the measurement. Parameter count and FLOPs are obtained using a dedicated model‑profiling utility to ensure consistency across architectures.
> Energy consumption is quantified using a hardware‑level power‑tracking tool that records instantaneous GPU power draw during the model’s forward pass and integrates this over time to obtain total energy in Joules. CO₂ emissions are then computed from the measured energy using the appropriate regional carbon‑intensity conversion factor. Memory usage is obtained from the process‑level resident memory footprint at inference. All measurements are performed under identical hardware and software conditions to ensure fair comparison across models.
> This clarification is included in the revised manuscript (see Section: Experimental Setup and Appendix:B) and highlighted in red.
>
>
> (4)  Response to the reviewer’s concern about undefined acronyms:
>
> Thank you for highlighting this omission. All acronyms used in the manuscript have now been defined at their first occurrence to ensure clarity and consistency for readers. The revised manuscript includes these additions, with the corresponding changes highlighted in red.
>
>
> (5) Response to the comment regarding Section 4 (discussion and conclusion):
>
> Thank you for this helpful suggestion. In accordance with the reviewer’s recommendation, the discussion and conclusion have been separated into two distinct sections in the revised manuscript, with the corresponding changes highlighted in red.

---

> > ### Comment · Reviewer_p2Ge · 2026-01-27
> > **Official comment**
> >
> > I would like to thank the author for addressing my concerns.

---

### Author Rebuttal · Authors · 2026-01-20

**Rebuttal:**

I would like to sincerely thank all the reviewers for their thoughtful, constructive, and encouraging feedback on this work. The comments provided were highly valuable and have helped strengthen both the clarity and the scientific rigor of the manuscript. In response to the reviewers’ suggestions, the manuscript has been carefully revised, and all modifications are highlighted in red for ease of inspection. The updated manuscript, along with the required supporting materials, has been uploaded as part of the rebuttal.

**Supporting Material:**

/attachment/5f3c1d349d4f7cca4c6c1e87e7a94818a54935d1.pdf

---

### Meta-Review · Area_Chair_sVHg · 2026-02-10

**Recommendation:** Accept (Poster)
**Confidence:** 4

**Metareview:**

This paper proposes QICNN, a quantum-inspired CNN that enforces orthonormal kernels and unitary channel mixing, yielding large reductions in parameters, latency and measured energy use while maintaining or improving denoising quality on brain MRI (GLCM metrics, CNR, and added PSNR/SSIM on phantom data). Two reviewers are very positive, highlighting a principled physics-guided design, clear writing, strong empirical gains, and promising clinical potential; the rebuttal further clarifies SVD overhead, energy/CO₂ computation, and adds stronger baselines (HAIR) plus phantom and CT experiments with statistical testing. A remaining borderline reviewer is still unconvinced about the choice to train on natural images and the limited modality/anatomy coverage relative to the “modality-agnostic” claim, and the quantum framing is more conceptual than genuinely quantum computing. Overall, however, the core idea (orthonormal/unitary constraints for efficient denoising) is technically sound, novel in this context, and supported by substantially improved efficiency and solid, if not yet exhaustive, validation.

---

### Decision · Program_Chairs · 2026-02-13

Accept (Poster)